# Research on the Parameter Test and Identification Method of Electromechanical Transient Model for PV Power Generation

**Tao Shi [1],\***[ID]**, Linan Qu [2] and Luming Ge [2]**

[1]  Institute of advanced technology, Nanjing University of Posts and Telecommunications,
    Nanjing 210023, China
[2]  China Electric Power Research Institute Co., Ltd., Nanjing 210003, China; qulinan@epri.sgcc.com.cn (L.Q.);
    geluming@epri.sgcc.com.cn (L.G.)
\*  Correspondence: shitao@njupt.edu.cn

**Abstract:** Model and parameters are the indispensable conditions for the simulation calculation of power systems with a high proportion of photovoltaic power generation. Conventional models of power electronic devices are difficult to meet the requirement of power system electromechanical transient simulation, and the parameters are difficult to obtain. Aiming at this problem, this paper proposes a structure of an electromechanical transient simulation model of a photovoltaic power station and designs a set of photovoltaic power generation transient characteristic test systems based on a fault simulation device. Through a disturbance test and model parameter identification, the electromechanical transient simulation model and parameters of photovoltaic power generation are obtained. In this paper, based on the test system, the electromechanical transient characteristics of a certain type of photovoltaic inverter are modeled. The results show that the model can successfully describe the electromechanical transient characteristics of photovoltaic power generation, and the simulation results obtained based on the model parameters have a good fitting degree compared with the measured curve.

**Keywords:** photovoltaic power generation; electromechanical transient; inverter; parameter identification

## 1. Introduction

Model parameters are the basis of power system simulation calculations. With the continuous maturity of photovoltaic (PV) power generation technology and its promotion and application in the world, the proportion of PV power generation in the installed capacity of power system continues to increase. The model parameters of PV power generation also become an indispensable condition for the simulation calculation of power systems with a high proportion of PV power generation. Many scholars have carried out extensive and in-depth research on PV power generation modeling requirements applicable to power system simulation calculation. In reference [1], according to the four parameters provided by the manufacturer under the standard test conditions, two detailed methods for calculating the model parameters are given. In reference [2], a disturbance observation method based on uncertainty reasoning was proposed to realize maximum power point tracking (MPPT). In reference [3], a MPPT technique combining a prediction model with a disturbance observation algorithm is proposed. In reference [4], an effective strategy is presented to realize IGBT open-circuit fault diagnosis for closed-loop cascaded PV grid-connected inverters. In reference [5], an overview of MPPT methods for PV systems used in the Micro Grids of PV systems is presented. In reference [6], a method to improve the performance of a distribution system by optimizing volt–var function of a

smart inverter to alleviate the voltage deviation problem due to distributed generation connection is proposed. In reference [7], a coordinated voltage and reactive power control architecture for large PV power plants is proposed. In reference [8], an improved simplified nonlinear engineering mathematical model of silicon solar cells is proposed. The output current of PV cells under any light intensity and temperature can be calculated by using the four electrical parameters under the standard test conditions provided by the PV cell manufacturer. In reference [9], the output characteristic curve of PV cells is proposed by using the trajectory of horizontal throwing. The test results show that the replacement model can meet the engineering application accuracy under the standard test conditions. Reference [10] is aimed at the problem that the transcendental model contained in the general model is difficult to solve, and a PV cell model based on the Bezier function is proposed. This method simplifies the modeling of PV cells based on the test data. The advantage of this method is that it does not need to estimate the internal parameters of the cell and is easy to implement.

With the improvement of accuracy requirements, many parameter identification methods based on intelligent algorithms [11–14] have been applied to photovoltaic single diode and dual diode equivalent circuit models. However, the PV model in the parameter identification method is usually limited to polynomial or transcendental functions in mathematics, which makes it difficult to optimize or solve the parameters, and also limits the adaptability of the model. Small external changes may lead to large parameter deviation, which cannot meet the flexibility requirements in many cases. Therefore, the data-driven modeling method based on machine learning and artificial intelligence technology has been gradually applied in PV modeling due to its few constraints, flexible structure, and strong adaptability. In reference [15], a white adaptive neural network MPPT control model is proposed to optimize the multi peak output characteristic curve. In reference [16], the relationship between temperature, light intensity, and maximum power point is established based on the back propagation (BP) network of fuzzy control. In reference [17], binary ant colony algorithm is used to optimize the parameters of the fuzzy neural network MPPT strategy. In reference [18], it is proposed to improve the convergence speed of the MPPT white adaptive neural network by load voltage ergodic feedback. In reference [19], it is proposed to use the convolutional neural network (CNN) to predict the probability of PV power for medium and long-term planning of power grids. In reference [20], a PV power probability prediction method based on Improved Bayesian neural network, taking into account the influence of persistence and suddenness, is proposed to improve the adaptability of short-term power prediction to accidental factors. Since the inverter, the core component of PV power generation, is a power electronic device and its interface characteristic response time scale is microseconds, the existing models mainly belong to the category of electromagnetic transient. The electromagnetic transient model of the inverter is very close to the actual physical process, considering the pulse width modulation and the conduction state of the inverter switch. It can simulate the high-frequency switching characteristics of the inverter, obtain accurate voltage and current transient waveform, and conduct harmonic analysis. Therefore, the electromagnetic transient model is very suitable for the design and analysis of inverter operation characteristics and control strategy. The simulation step size of the electromagnetic transient simulation is generally less than 100 µs, while the step size of the electromechanical transient simulation is generally 5–20 ms. After the inverter is filtered, most of the high frequency harmonics are filtered out, which has little impact on the power system. Therefore, it is unnecessary to consider such a detailed switching process in electromechanical transient simulation. However, the large-scale power system has a complex network and a large amount of equipment. A single PV power station contains hundreds of inverters. The direct use of the microsecond-level model will reduce the simulation efficiency of the electromechanical transient of the power system, and the measured parameters are difficult to obtain. Therefore, this paper aims to establish a PV power generation model suitable for the electromechanical transient simulation of power system and proposes a test and identification method for its model parameters. Through a disturbance test and parameter identification based on the test system, the model and parameters of the electromechanical transient model of PV power generation can be successfully fitted.

## 2. Electromechanical Transient Model Structure

This paper mainly studied the mathematical model of the PV power station suitable for electromechanical transient simulation of the power system. Dynamic processes such as maximum power point tracking (MPPT) of inverter and switch tube modulation are not considered in the electromechanical transient modeling process of the PV power station. In the process of standardization of the parameters of the PV power generation unit, the reference capacity should be the total rated power of the inverter in the content of the PV power generation unit, and the reference voltage on the Direct Current (DC) side should be the standard working condition (solar irradiance $S_{ref}$ = 1000 W/m², PV array working temperature $T_{ref}$ = 25 °C) under the working voltage of the maximum power point of PV array, the reference voltage on the alternating current (AC) side should be the rated voltage of the PV power generation unit boosted to the low voltage side.

### 2.1. Typical Structure of the PV Power Station

The PV power station consists of several PV power generation units, collector lines, step-up transformers in the station, reactive power compensation devices in the station, etc. The PV power generation unit is composed of PV array, inverter, cell step-up transformer, and so on. Specifically shown in Figure 1.

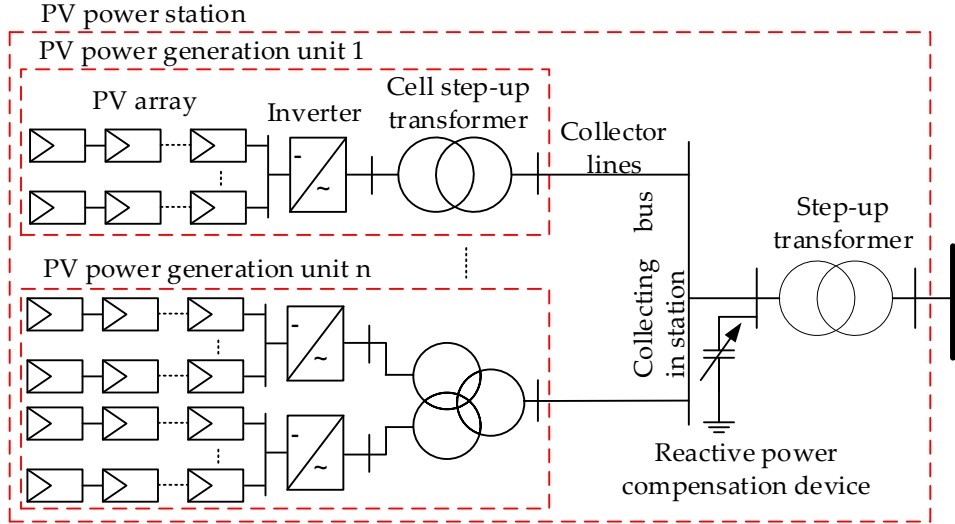

**Figure 1.** Typical structure of a photovoltaic (PV) power station.

Corresponding to the physical structure of the PV power station in Figure 1, the mathematical model of the PV power station is also composed of the PV power generation unit model and conventional power equipment element model [21]. The model parameters of the PV power generation unit should be obtained by means of actual measurement, and PV power generation units composed of PV square arrays and inverters of different types and capacities should be modeled, respectively.

### 2.2. Model Framework

The electromechanical transient calculation model of the PV power station is composed of multiple PV power generation unit models and conventional power equipment element models, as shown in Figure 2. The electromechanical transient model of the PV power generation unit consists of three parts [22]: (1) PV array model: simulates the nonlinear characteristics of PV array; (2) inverter model: simulates the grid-connected operation control characteristics of the inverter, including grid-connected interface models, control and protection models, etc.; (3) unit step-up transformer model. Among them, the inverter model is the core of electromechanical transient modeling.

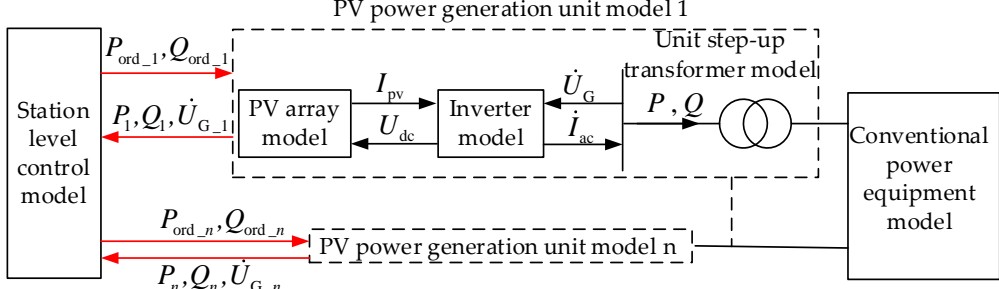

**Figure 2.** Electromechanical transient model framework.

## 2.3. PV Array Model

The PV array power output model simulates the photoelectric conversion characteristics of PV array under different environmental factors. The environmental quantity input is the solar irradiance S of the current working condition and the working temperature T of the current working condition. The input electrical quantity is the DC working voltage $U_{dc}$ of the PV array. The electrical quantity output is the PV array output current $I_{pv}$. According to the test parameters under standard test conditions, the characteristics of the PV array IV under any radiation intensity S and operating temperature T can be derived [8].

## 2.4. Inverter Control Model

The connection relationship of the three-phase grid-connected PV inverter is shown in Figure 3:

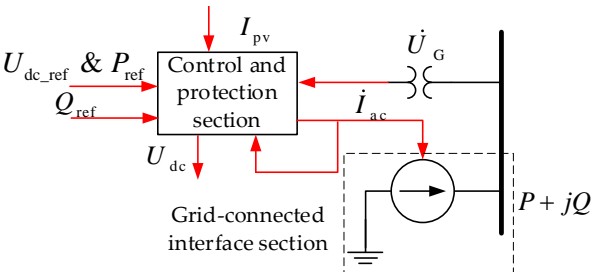

**Figure 3.** Connection relation of electromechanical transient model of inverter.

The output apparent power $\widetilde{S}$ of the inverter is [23]:

$$\widetilde{S} = \dot{U}_G \cdot \dot{I}_{ac}^* = P + jQ \tag{1}$$

The inverter control and protection model includes the steady-state operation control module and the transient control protection module, as shown in Figure 4.

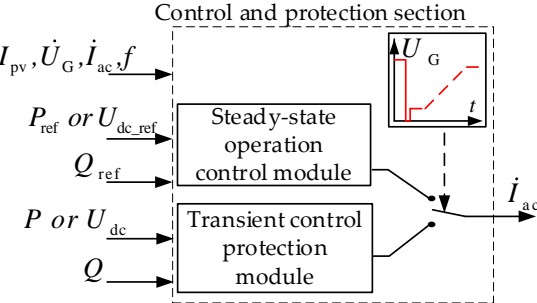

**Figure 4.** Inverter control and protection model.

In steady-state operation, the inverter adopts active and reactive decoupling control, including DC voltage calculation module, active power control module, reactive power control module and output current calculation module, as shown in Figure 5.

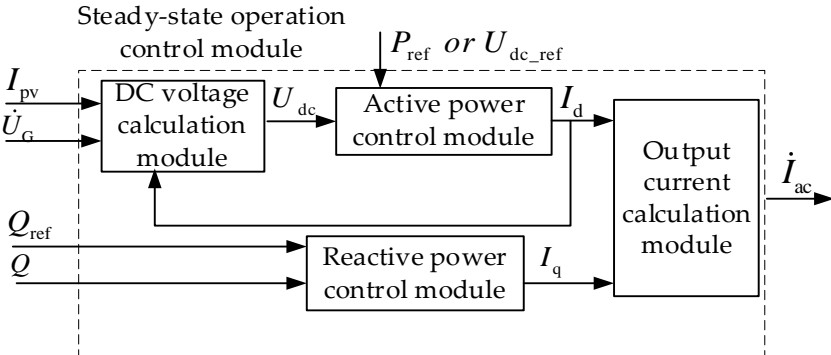

**Figure 5.** Steady-state operation control module of inverter.

The DC voltage calculation module simulates the stable voltage of the DC capacitor of the inverter. The active power control module tracks the maximum power point voltage (or control command at plant-level) of the PV array, and outputs the inverter current active component $I_d$. The block diagram of the active power control module with $U_{dc}$ as the controlled object is shown in Figure 6.

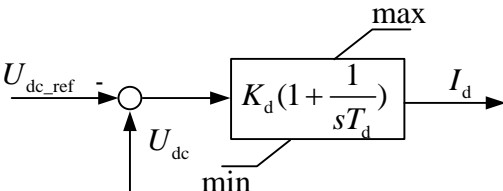

**Figure 6.** Block diagram of active power control module.

The reactive power control module takes the reference value of reactive power as the control target and outputs the inverter current reactive component $I_q$, as shown in Figure 7.

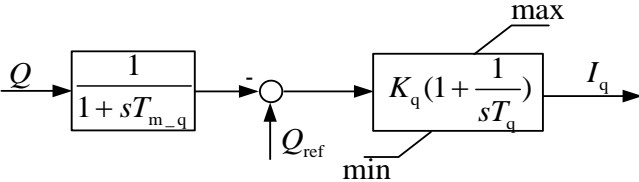

**Figure 7.** Block diagram of reactive power control module.

The output current calculation module calculates the output current vector of the inverter according to the current active component, reactive component, and grid voltage phase [23].

$$\dot{I}_{ac} = \left( \frac{|\dot{U}_G| \cdot I_d - j |\dot{U}_G| \cdot I_q}{\dot{U}_G} \right)^* \tag{2}$$

According to the requirements of GB/T 19,964 for the fault ride-through and recovery process of the PV power station, the dynamic reactive current injected into the grid by PV power station should meet certain requirements, and the active power restores to the pre-fault value at a certain rate after fault clearing.

The variables and parameters of inverter protection module as shown in Figure 8, usually include control node voltage class, voltage, current, frequency limit, tolerance time, protection action time, etc., which can be used as a reference value according to the inverter setting value.

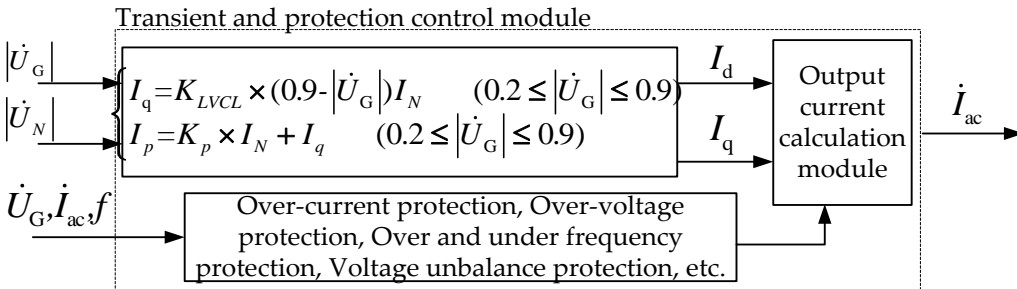

**Figure 8.** Block diagram of transient and protection control module.

The output current calculation module refers to the definition in the steady-state operation control module.

## 3. Transient Characteristic Test

### 3.1. Test Principle

The test principle of transient characteristic of PV power station is shown as in Figure 9.

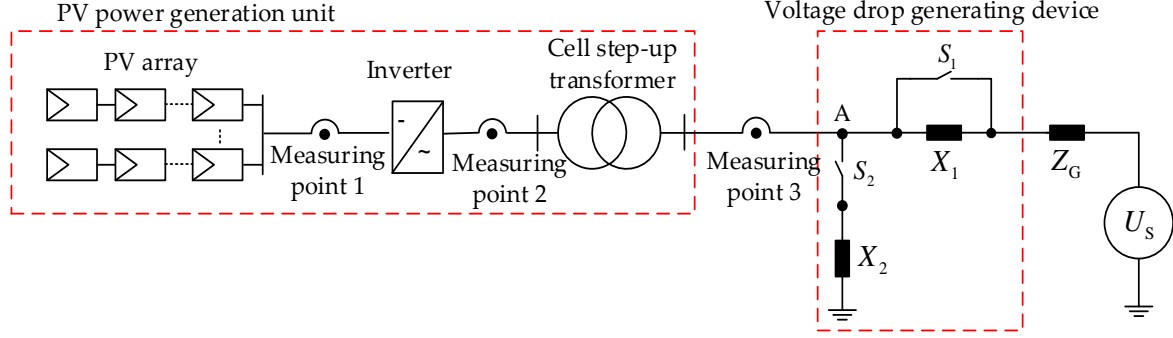

**Figure 9.** Schematic diagram of model parameter test.

### 3.2. Verification Conditions

(1) Power range: high power output state, P ≥ 0.8 Pp, P is the output active power of the PV power generation unit, Pp is the peak power of the PV power generation unit; intermediate power grade output state, 0.5 Pp ≤ P ≤ 0.7 Pp; low power output state, 0.1 Pp ≤ P ≤ 0.3 Pp.

(2) Fault type: the working conditions of the electromechanical transient model parameters of the PV power generation unit include a three-phase symmetric fault and a two-phase asymmetric fault.

(3) Test data: the instantaneous values of voltage and current on the DC side and AC side of the inverter are measured synchronously.

### 3.3. AC Small Disturbance Test

(1) The model parameter test should be selected when the output of the inverter is within the high power output range;

(2) exit the inverter MPPT control mode and enter the constant voltage control mode. The reference value of DC voltage is set to 1 p.u. to stable operation;

(3) set the voltage drop (or rise) of the AC side of the inverter to 0.95 and 0.9 p.u. (or 1.05 and 1.1 p.u.), respectively, and the stable operation of the inverter lasts 2 s;

(4)    restore the AC side voltage of the inverter to 1 p.u. until stable operation of the inverter.

*3.4. AC Large Disturbance Test*

(1)    The model parameter test should be carried out when inverter output is within the high power output range;

(2)    exit the inverter MPPT control mode and enter the constant voltage control mode. The reference value of DC voltage is set to 1 p.u. to stable operation;

(3)    set the voltage drop of the AC side of the inverter to 0.85 p.u. until stable operation of the inverter, and set the voltage drop of the AC side of the inverter to 0.6, 0.4, 0.2, and 0 p.u. until stable operation of the inverter;

(4)    restore the AC side voltage of the inverter to 0.85 p.u. until stable operation of the inverter;

(5)    restore the AC side voltage of the inverter to 1 p.u. until stable operation of the inverter.

## 4. Model Parameter Identification

*4.1. Preprocessing of Test Data*

(1)    Low-pass filter: due to the influence of the on-site test environment, the test data contained a large number of harmonics, which needed to be filtered. In this paper, a first-order low-pass filter was selected. The selection of filter parameters affects the amplitude and phase angle of the measured data. Therefore, the amplitude and phase angle difference caused by the filter must be considered when calculating the output power of the PV power generation unit according to the filtered voltage and current.

(2)    Extract the positive sequence component of the fundamental wave: the object of parameter identification is the effective value of the fundamental positive sequence component of each parameter. Therefore, after filtering the high-order harmonic components, extract the fundamental positive sequence components of the voltage and current, then calculate the active power and reactive power output of the PV power generation unit.

(3)    Power calculation: the active power and reactive power output of the PV power generation unit are calculated according to the filtered instantaneous values of voltage and current. Among them, the phase difference between voltage and current needs to compensate the phase difference between line voltage and phase current and the phase difference caused by the filter. At the same time, the module calculates the effective values of voltage and current.

(4)    Reduce the sampling rate: to ensure the validity of the comparison between test data and simulation data, all data should be in the same dimension, time scale, and resolution format. The disturbance test data are the instantaneous values, and the previously processed data still have a high sampling rate (for example, 100 kHz), while the simulation step of power system transient calculation is millisecond class (for example, 20 ms). Therefore, the average value of the test data is calculated within a simulation step to obtain the test comparison data, and the test data are processed to per unit values.

*4.2. Identification Method*

According to the theory of system identification, identification methods can be divided into a classical identification method, modern identification method, and artificial intelligence identification method. Classical identification methods mainly include a convolution identification method, correlation identification method, and frequency domain FFT method etc. [24]. Modern identification methods mainly include a least square estimation method, maximum likelihood method, and Kalman filtering method, etc. Typical intelligent identification methods include genetic algorithm and immune

algorithm. In this paper, the least squares estimation method was adopted [25], which is based on the minimum sum of squares of the difference between the measured value $z$ and the estimated value $\hat{z}$.

$$h_i(x) = \sum_{j=1}^{n} h_{ij}x_j, \ i = 1, 2, \cdots, m \tag{3}$$

$$z = Hx + v \tag{4}$$

where $x$ is the true value of the state variable; $n$ is the number of state variables; $h(x)$ is the measurement function; $m$ is the number of measurements; $H$ is a matrix $m \times n$, and its element is $h_{ij}$.

The objective function is established according to the least squares criterion [25]:

$$J(x) = (z - Hx)^T(z - Hx). \tag{5}$$

The estimator $\hat{x}$ can be solved by differentiating the objective function and setting it to zero [25].

$$\frac{\partial J(x)}{\partial x} = 0 \tag{6}$$

## 5. Model Validation

### 5.1. Model Simulation

The electromechanical transient model of the PV station was established in power system software. Before model simulation, the power system parameters included voltage, equivalent impedance, and short circuit capacity should be determined.

### 5.2. Model Evaluation

According to the time scale, the test and simulation data were processed in sections. Taking a large disturbance test on the AC side as an example, the data were divided into three sections [26].

- Period A: before fault, 2 s before voltage disturbance is the beginning of period A, and the beginning time of voltage disturbance is the end of period A.
- Period B: during the fault period, the end of period A is the beginning of period B, and the start of voltage disturbance clearing is the end of period B.
- Period C: after the fault, the end of period B is the beginning of period C, and the 2 s after the active power of PV unit starts to output stably is the end of period C.

$Xs$ and $Xm$ were, respectively, used to represent the per-unit values of electric quantities provided by model simulation and experimental test. The sequence numbers of the first and last data obtained from the model simulation or experimental test in the corresponding interval are represented by $K_{S\_start}$, $K_{M\_start}$, $K_{S\_end}$, and $K_{M\_end}$, respectively. The deviation indexes of each interval were as follows [26]:

- Average deviation of the steady state interval: F1.

$$F_1 = |\frac{1}{K_{S\_End} - K_{S\_Start} + 1} \sum_{i=K_{S\_Start}}^{K_{S\_Start}} X_s(i) - \frac{1}{K_{M\_End} - K_{M\_Start} + 1} \sum_{i=K_{M\_Start}}^{K_{M\_Start}} X_M(i)| \tag{7}$$

- Average deviation of the transient interval: F2.

$$F_2 = |\frac{1}{K_{S\_End} - K_{S\_Start} + 1} \sum_{i=K_{S\_Start}}^{K_{S\_Start}} X_s(i) - \frac{1}{K_{M\_End} - K_{M\_Start} + 1} \sum_{i=K_{M\_Start}}^{K_{M\_Start}} X_M(i)| \tag{8}$$

- Maximum deviation of the steady state interval: F3.

$$F_3 = max_{i=K_{M\_End}-K_{M\_Start}} (| X_s(i) - X_M(i) |) \tag{9}$$

The weighted average total deviation of voltage, current, active power, and reactive power in the whole disturbance process were obtained by calculating the average deviation of each period. In this paper, the weights of the three intervals were 10% (period A), 60% (period B), 30% (period C).

## 6. Case Study

Based on the above research, a set of test platforms was designed, as shown in Figure 10. The system used a controllable DC power supply to simulate the V-I input characteristics of the PV array and connected the DC side of the inverter. The AC side of the inverter was connected to the power grid fault simulation device through a step-up transformer. The power grid fault simulation device connected/disconnected reactor $X_1$ and $X_2$ through circuit breakers S1 and S2 to simulate various fault disturbances of the power grid voltage. In order to reduce the impact on the power grid, a buffer reactance $Z_G$ was connected in a series between the grid fault simulation device and the grid. Two measuring points were set on the DC side and AC side of the inverter to collect DC side voltage, DC side current, AC side voltage, and AC side current. The measured data were used to identify the model parameters of the PV inverter.

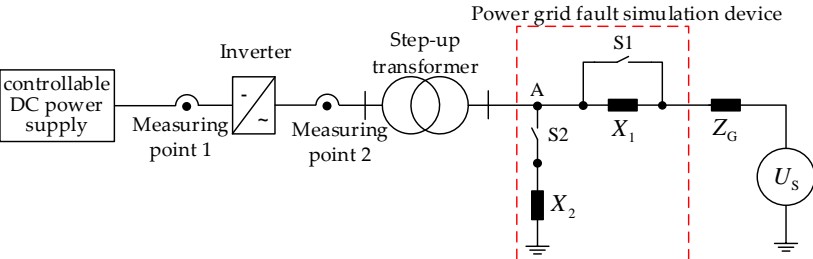

**Figure 10.** Schematic diagram of the test system.

The test items included a voltage small disturbance test and a voltage large disturbance test. Among them, the small disturbance test referred to the use of the power grid fault simulation device to change the grid-side voltage, the voltage range was 0.9~1.15 times the rated voltage, and tested whether the inverter could operate stably in the case of the small disturbance of the grid voltage. The large disturbance test meant that the grid side voltage dropped to 0~0.9 times the rated voltage, and it tested whether the inverter could maintain grid-connected operation and provide reactive current support to the grid when a serious short circuit fault occurred in the grid.

### 6.1. Parameter Identification under Small Disturbance Mode of the Grid Voltage

The rated power of the tested PV inverter was 500 kW, the grid-side rated line voltage was 315 V, and the working range of the DC side voltage was 500~850 V. Test device parameters: the short-circuit capacity of the power grid fault simulation device was 2 MVA, and the reactance of the generating device was $L_1 + L_2 = 200$ mL; the rated capacity of the transformer was 2 MVA with a ratio of 10.5/0.315 kV, and the short-circuit voltage was 4%. The test steps were as follows:

(1) Make the output of the inverter reach more than 0.7 p.u. of its rated output power;
(2) After the inverter enters into steady-state operation, the circuit breaker S2 is closed through the grid fault simulation device, and makes the AC side voltage of the inverter drop to near 0.925 p.u.;
(3) The test data are the instantaneous values of DC voltage at measuring point 1 and AC voltage and current at measuring point 2. The voltage and current transformers are used to transmit the

measurement signal to the multi-channel wave-recorder to ensure the synchronization of the test data of each channel in time.

The waveform of the grid voltage is shown in Figure 11. Since the unit power factor control was adopted in the test inverter, reactive power was not output in a normal operation and the control reactive current was 0, so there was no need to identify the parameters of the reactive controller. According to the grid voltage disturbance data, the data of the controller were identified. The identification results are shown in Table 1, and the fitted curve of the active current is shown in Figure 12.

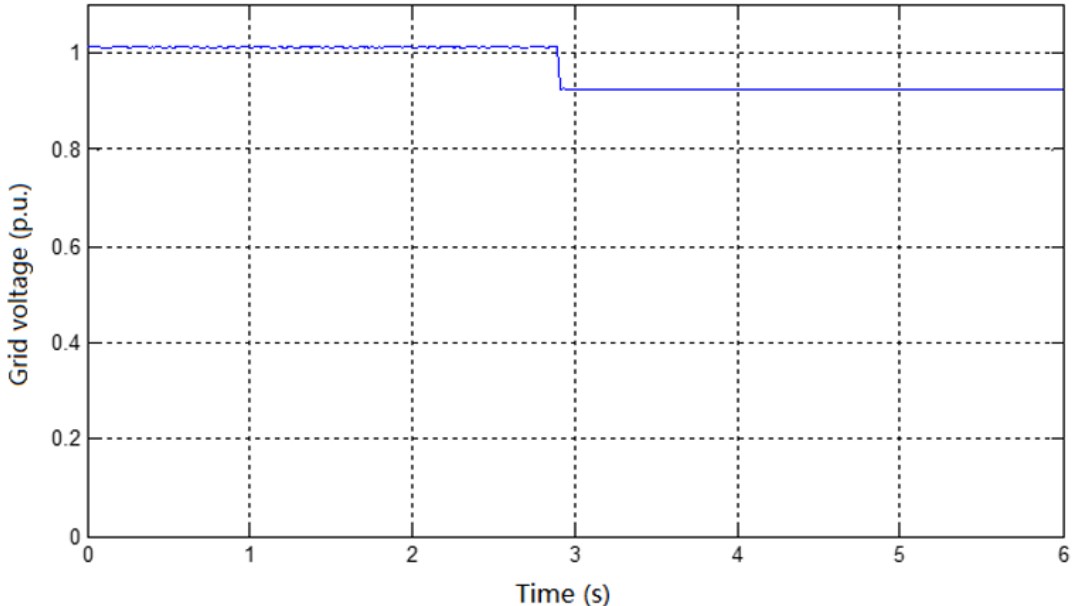

**Figure 11.** Grid voltage curve with small disturbance of grid voltage.

**Table 1.** Parameter identification results under small disturbance of grid voltage.

| Identification Parameters | Damped Least Squares |
|:---:|:---:|
| $K_d$ | 0.00269 |
| $T_d$ | 0.11447 |

**Figure 12.** Active current fitted curve under small disturbance of grid voltage.

The identification curve shows that the active current simulation curve could fit the measured curve closely under the small disturbance of the grid voltage, and the identified parameters were also within a reasonable range, as shown in Table 2. The identification results verified the correctness and validity of the above theoretical analysis in the previous section.

**Table 2.** Model evaluation of active current under small disturbance of grid voltage.

| Period | Deviation Type | Deviation Value |
|---|---|---|
| Period A | Average Deviation | 0.002178 |
|  | Maximum Deviation | 0.0048 |
| Period B | Average Deviation | 0.000343 |
| Period C | Average Deviation | 0.000492 |
|  | Maximum Deviation | 0.0030 |
| Whole Process | Weighted Average Deviation | 0.000571 |

*6.2. Parameter Identification under Large Disturbance Mode of the Grid Voltage*

The test steps and data measuring points were the same as above. During the test, the voltage dropped to 0.63 p.u. of the rated voltage. The waveform of grid voltage is shown in Figure 13.

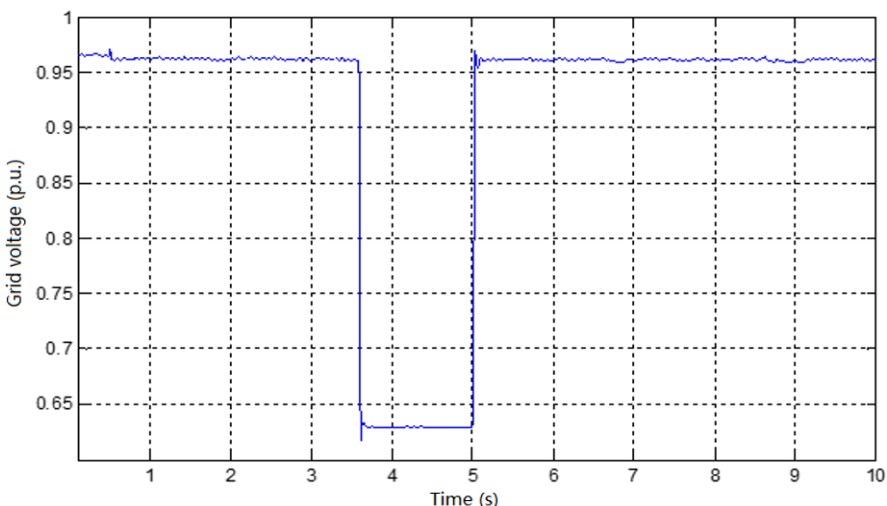

**Figure 13.** Grid voltage curve with large disturbance of grid voltage.

According to the PV inverter fault ride-through control model mentioned above, the model parameters determined by the test data are shown in Table 3, and the fitting curves of active and reactive power current obtained are shown in Figures 14 and 15.

**Table 3.** Parameter identification results under large disturbance of grid voltage.

| Identification Parameters | Damped Least Squares |
|---|---|
| $K_{LVLC}$ | 1.55 |
| $K_P$ | 1.05 |

The identification curve shows that the active current component and reactive current component of the inverter output could fit the measured curve closelyunder the large disturbance of the grid voltage, and the identification parameters were within a reasonable range, as shown in Tables 4 and 5, which verified the correctness and effectiveness of the above test and identification methods.

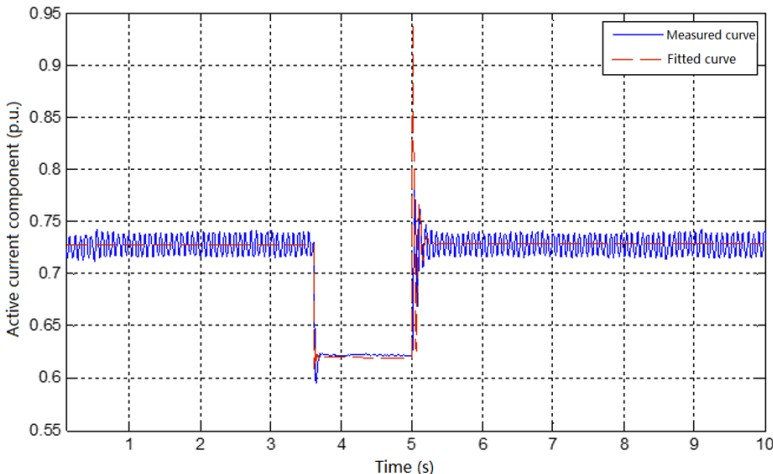

**Figure 14.** Active power fitted curve of large disturbance of grid voltage.

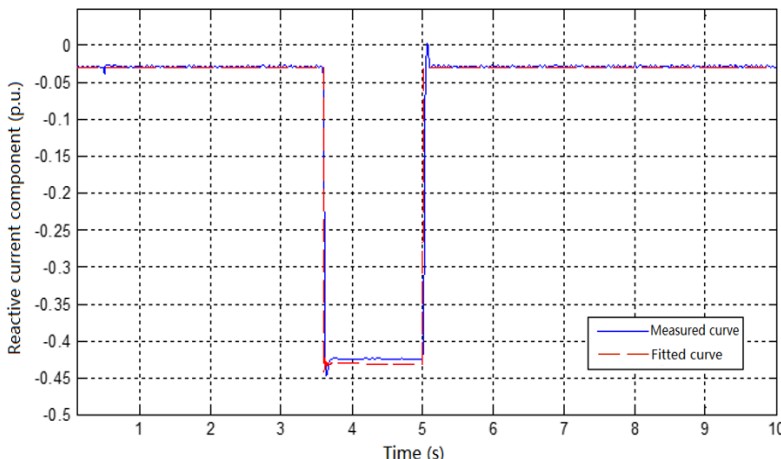

**Figure 15.** Reactive power curve of large disturbance of grid voltage.

**Table 4.** Model evaluation of active current under large disturbance of grid voltage.

| Period | Deviation Type | Deviation Value |
|---|---|---|
| Period A | Average Deviation | 0.000813 |
| | Maximum Deviation | 0.001898 |
| Period B | Average Deviation | 0.06815 |
| Period C | Average Deviation | 0.03441 |
| | Maximum Deviation | 0.314974 |
| Whole Process | Weighted Average Deviation | 0.051292 |

**Table 5.** Model evaluation of active current under large disturbance of grid voltage.

| Period | Deviation Type | Deviation Value |
|---|---|---|
| Period A | Average Deviation | 0.0004 |
| | Maximum Deviation | 0.016228 |
| Period B | Average Deviation | 0.005467 |
| Period C | Average Deviation | 0.00156 |
| | Maximum Deviation | 0.028566 |
| Whole Process | Weighted Average Deviation | 0.003788 |

## 7. Conclusions

This paper mainly studies the PV power station model suitable for electromechanical transient simulation of a power system. The research shows that the transient characteristics of PV power generation are mainly affected by the inverter control strategy and interface characteristics. Therefore, the inverter model is the core of the whole modeling work. In this paper, the electromechanical transient model structure of the PV inverter is established based on the time scale of electromechanical simulation and the control protection strategy of the PV inverter, and a model parameter identification method based on the transient characteristic test and test data fitting is proposed. Based on a certain test system, the test and parameter identifications are carried out in small and large disturbances of the grid voltage, respectively, in the paper. The results show that the parameters of the PV power generation model obtained through the test and identification can successfully fit the electromechanical transient characteristics, and have certain engineering practicability. In particular, it can provide basic models and parameters for power system security and stability simulation and control decision-making, including PV power stations, and improve the simulation efficiency on the premise of ensuring the accuracy of simulation results.

**Author Contributions:** This research was a collaborative effort between the authors. T.S. proposed the method framework of PV parameter identification, designed the experiments and wrote the paper. L.Q. realized the simulation model of the PV system in the software and performed the experiments. L.G. contributed analysis tool and reviewed the paper. All authors have read and agreed to the published version of the manuscript.

**Funding:** This research received no external funding.

**Conflicts of Interest:** The authors declare no conflicts of interest.

## Nomenclature

| Symbol | Definition | Unit |
|--------|-----------|------|
| $S_{ref}$ | Light intensity corresponding to standard test conditions | W/m$^2$ |
| $T_{ref}$ | Ambient temperature corresponding to standard test conditions | °C |
| $P_{ord}$ | Active power control command of PV power station | kW |
| $Q_{ord}$ | Reactive power control command of PV power station | kvar |
| $P$ | Active power output of PV inverter | kW |
| $Q$ | Reactive power output of PV inverter | kW |
| $U_G$ | Interface voltage of PV inverter | V |
| $U_{dc}$ | DC voltage of PV inverter | V |
| $I_{pv}$ | Current output of PV array | A |
| $I_{ac}$ | Current output of PV inverter | A |
| $U_{dc\_ref}$ | DC reference voltage of PV inverter | V |
| $P_{ref}$ | Active power reference of PV inverter | kW |
| $Q_{ref}$ | Reactive power reference of PV inverter | kvar |
| $I_d$ | Active current component of PV inverter | A |
| $I_q$ | Reactive current component of PV inverter | A |
| $K_d$ | Proportional coefficient of active current control | |
| $T_d$ | Time constant of integral link of active current control | s |
| $K_q$ | Proportional coefficient of reactive current control | |
| $T_q$ | Time constant of integral link of reactive current control | s |
| $T_{m\_q}$ | Inertia time constant of reactive current control | s |
| $U_N$ | AC rated voltage of PV inverter | V |
| $I_N$ | AC rated current of PV inverter | A |
| $K_{LVCL}$ | Active current support coefficient of inverter in fault ride through | |
| $K_p$ | Reactive current support coefficient of inverter in fault ride through | |

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
