# Peer review of "Research on the Parameter Test and Identification Method of Electromechanical Transient Model for PV Power Generation"

_electronics, doi:10.3390/electronics9081184_

Round 1
Reviewer 1 Report
The topic is interesting and it is adapt to this journal. The collaboration among several faculties is useful and I think that there is a great work behind the presentation of this work. However, while the presentation is nice in shape, there are few comments and/or suggestions to improve the manuscript.
-According to scientific standards, abbreviations cannot be used in the abstract, please correct it in the manuscript.
-Clarify better the innovation of this work in the abstract and in the main text.
-Read articles to understand the structure of Electronics. The following structure would be preferable based on the Sensors Microsoft Word template file: 1. Introduction (1.1, 1.2, 1.3.), 2. Materials and Methods (2.1, 2.2., 2.3.), 3. Results and Discussion (3.1, 3.2, 3.3), 4. Conclusions. These sections mixed in the text. The introduction section is a literary review of the topic. In the introduction (or where still necessary) all paragraph must be cited because of the risk of plagiarism.
https://www.mdpi.com/journal/electronics/instructions
-A short paragraph introducing the problem statement and actions taken (or a description of the study) should be included at the end of the Introduction section.
-It would be important to compare the results with other „new” modeling concepts:
-Please add more references because the number of scientific references is low. Please provide more general information on the importance of this topic in the introduction. At least 25-30 scientific manuscript need to use because this is a Q2 Journal! What is the role of this topic in literature and in international context?
-Please search references to the equations. Equations should always be accurately and clearly referenced.
-Please add more information's about the model validation in a new chapter. Please better support the accuracy of the model with measurement results.
The manuscript must be at least 15-20 pages long.
-Extend the conclusion with more general usability. What are the benefits of the results in a global context? Please explain this better in the manuscript.
-At the end of the study need to create a nomenclature table with units.
Author Response
Dear expert:
Thank you for your review. I have revised the paper according to your suggestion, and the details are shown in the attached file.
Kind regards!

Reviewer 2 Report
The paper concerns photovoltaic power generation, which is a timely topic. The writing is a bit repetitive at places. The math is pretty standard. Overall, I think the paper is acceptable.
A few comments:
Abstract: “With the increasing proportion of photovoltaic (PV) power generation in the installed 10 capacity of the power system, its model parameters become an indispensable condition for the simulation calculation of power systems with high proportion of PV power generation.” Note that the phrase ‘proportion of photovoltaic (PV) power generation’ is used twice in this sentence and the word ‘power’ appears 4 times. This makes the sentence a bit difficult to read. Consider breaking it up into two sentences or rewriting it.
Introduction: “Domestic and foreign scholars” What is the meaning of this distinction in a journal with an international readership.
The equations that are typeset inside the text could look nicer (use spaces and italic font).
There are not many references, but those provided are very current.
Author Response

(The authors gave the same response as above.)
